# Optimal Endoscopic Resection Technique for Selected Gastric GISTs. The Endoscopic Suturing System Combined with ESD—a New Alternative?

**DOI:** 10.3390/jcm9061776

**Published:** 2020-06-08

**Authors:** Katarzyna M. Pawlak, Artur Raiter, Katarzyna Kozłowska-Petriczko, Joanna Szełemej, Jan Petriczko, Katarzyna Wojciechowska, Anna Wiechowska-Kozłowska

**Affiliations:** 1Department of Internal Medicine, Cardiology, Gastroenterology and Endocrinology, Hospital of the Ministry of Interior and Administration, 70-382 Szczecin, Poland; katjawoj@o2.pl (K.W.); annamwk@wp.pl (A.W.-K.); 2Department of Endoscopy, Specialist Hospital of Alfred Sokolowski, 58-309 Wałbrzych, Poland; artur.raiter@outlook.com (A.R.); joannaszelemiej@gmail.com (J.S.); 3Department of Gastroenterology and Internal Medicine, SPWSZ Hospital, 70-382 Szczecin, Poland; kasia-petriczko@outlook.com; 4Department of Plastic, Endocrine and General Surgery, Pomeranian Medical University, 70-382 Szczecin, Poland; jan.petriczko@gmail.com

**Keywords:** GISTs, EFTR, ESD, endoscopic suturing system, endoscopic resection

## Abstract

**Background and Study Aim: **In terms of therapeutic management, gastrointestinal stromal tumors (GISTs) seem to be the most difficult group of subepithelial gastrointestinal lesions (SELs). Despite various treatment option, choice of optimal management remains a dilemma in daily practice. Our aim was to evaluate a new hybrid resection technique of gastric GISTs type III as a modality of endoscopic full-thickness resection. **Methods: **Three males and one female (mean age of 68) were qualified for the procedure. Endoscopic full-thickness resections consisted of the endoscopic resection combined with suturing by Apollo OverStitch System. The main inclusion criterium was a complete diagnosis of GISTs (computed tomography (CT), endoscopic ultrasound (EUS), fine-needle biopsy (FNB)) with the evaluation of the tumor features, especially, the location in the gastric wall. All of the tumors were type III with a diameter between 20–40 mm. The lesions were located in the corpus (1), antrum (1) and between gastric body and fundus (2). All procedures were performed in 2019. **Results: **The technical and therapeutic success rate was 100% and the mean resection time 107.5 min. Neither intra- nor postprocedural complications were observed. In all four cases, R0 resection was achieved. Histopathologic assessment confirmed GIST with <5mitose/50HPF in all of the tumors, with very low risk. **Conclusion**: Based on our outcomes, endoscopic resection combined with the sewing by Apollo OverStitch of gastric GISTs type III, with the diameter between 20–40 mm, seems to be an effective therapeutic option with a good safety profile, however further studies with a larger treatment group are needed.

## 1. Background

In terms of therapeutic management, gastrointestinal stromal tumors (GISTs) seem to be the most challenging group of subepithelial gastrointestinal lesions (SELs). Despite various treatment strategies considering general clinical assessment and the type of tumor according to Miettinen classification, chose of optimal management in daily practice remains a dilemma for some group of tumors. After the confirmation of malignancy potential, regular follow-up for most GISTs with the diameter below 10 mm is widely accepted. Nevertheless, in case of GISTs with the diameter > 10 mm, R0 resection should be considered [1,2,3,4,5,6,7].

Endoscopic ultrasound (EUS) with immunohistochemical and pathologic assessment is a crucial method characterized by high sensitivity and specificity for the differential diagnosis of GISTs [8,9,10]. However, recent studies show that in 40.6% performed biopsies (19 and 22 G needles) obtained material was sufficient for evaluation the mitotic count on 50 HPF [11,12,13,14]. Considering the mitotic index as a malignancy predictor and its clinical implication, a complete resection of the tumor allows for the estimation of malignant potential and healing.

Presently, the first-line treatment for local GISTs is surgical resection [15,16,17]. In recent years, along with the development of advanced endoscopic techniques, new therapeutic pathways have been introduced for tumors with a diameter of < 40 mm. Recent studies have demonstrated feasibility and safety of endoscopic resection for tumors with the diameter below 50 mm [18]. Endoscopic management may be considered after complete diagnosis claimed by imaging examinations, pathologic result and after exclusion of high-risk factors [19]. Among the endoscopic resection modalities of GISTs located in the stomach were found endoscopic band ligation (EBL), endoscopic submucosal excavation (ESE), endoscopic submucosal dissection (ESD) and various combination of endoscopic full-thickness resection (EFTR) [20]. Submucosal tunneling endoscopic resection (STER) is the treatment option, though reserved for gastric GISTs localized mainly in the cardia. The selection of endoscopic techniques should consider the connection with muscularis propria (MP), size of the tumor and relation to individual layers of the stomach wall. The classification presented by Kim et al. [7] concerning the endoscopic/surgical approach depends on the connection with muscularis propria, may be useful in a choice of resection technique (Figure 1).

According to Kim et al. [7] classification, type I has a narrow connection with the MP layer, therefore the best therapeutic option for gastric GISTs are: EBL, ESE and ESD. Type II has a wider connection with muscularis propria and may be resected the same technique as type I. In the case of type III and IV, the achievement of complete resection by endoscopic methods is nearly impossible. Therefore, as well as EFTR techniques and surgical treatment should be considered for both of these types [7].

In current study, we present a results of combined endoscopic resection technique for type III of gastric GISTs.

## 2. Materials and Methods

### 2.1. Patients Selection and Tumors Characteristic

The selection of patients was based on the evaluation of particular clinical features and optimal care of patients after multidisciplinary committee qualification. All procedures were performed in 2019 at two Endoscopic Units (Hospital of the Ministry of Interior and Administration Szczecin, Poland and Specialist Hospital of Alfred Sokolowski, Wałbrzych, Poland). Every patient signed a written consent and was thoroughly informed about the course of the treatment. All the patients included into the research fulfilled the following criteria:(1)disqualification from surgical treatment due to concomitant diseases or lack of consent for surgical treatment,(2)gastric GIST with a diameter of >20 and <40 mm, evaluated with endoscopic ultrasound and confirmed by EUS-guided biopsy.(3)confirmed the local disease, without metastases or infiltration of local tissues (CECT, EUS).

The procedures were performed in a total of four patients (3 males and 1 female) at the mean age of 68 (62–77). In summary, three of the patients did not agree with surgical management. One patient was disqualified due to thrombocytopenia related to myelodysplastic syndrome. All patients underwent endosonographic evaluation of the tumor and surrounding tissues. Among the significant features of the were assessed: size and morphology of the tumor, location in the gastrointestinal wall and connection with particular layers (Figure 2). All EUS examination were performed by A.W-K. with the Pentax linear echoendoscope (processor Hi Vision Preirus, Hitachi Aloka Medical, Wallingford, CT). The tumors were located in the middle of the gastric body (1), in the antrum (1) and between the body and fundus of the stomach (2), with a mean size of 28.75 mm (20.0–40.0 mm). The deep connection with the muscularis propria was confirmed and the tumors were located in the middle of the gastric wall (type 3 according to Kim et al. classification). EUS-guided fine-needle biopsy (FNB; type of the needle—Expect™ 19 Flex; Boston Scientific, Natick, MA, USA) was performed for the pathologic conformation and assessment of immunohistochemistry. Selected markers like: SMA (-), DOG-1 (+), CD 117 (+), CD 34 (+) and Vimentin (+) were determined. Relevant data of the patients and lesions were described in Table 1.

### 2.2. Description of the Endoscopic Resection Procedure

All procedures were performed under general anesthesia in a supine position. For the standard endoscopic resection as a first step of the procedure, endoscope GIF-HQ 190 was used. In turn, for the suturing, endoscope GIF 2TH180 (Olympus America, Center Valley, Penn) was applied. All resections were performed by A.R. The procedure began with the injection of indigo carmine solution underneath the mucosal layer and circular incision of the mucosa around the tumor using the Dual Knife (Olympus, Tokyo, Japan) (Figure 3A–B). The next step was the dissection of submucosa. Then, in order to prevent perforation at the cutting site, Apollo OverStitch FM (Austin TX) was used to duplicate the gastric wall below the tumor (Figure 3C–D; Figure 5A–B). Duplication consists of doubling the whole gastric wall just underneath the tumor, which is possible by making them close together, through the sutures (2.0 continuous suture). This leads to the tumor elevation on duplicated folds. The duplication using Apollo OverStitch is starting at the site of previously cut submucosa. After the first duplication of MP, further cutting the muscularis propria above the sutured doubled gastric wall up was performed (Figure 4A–D). When the site below the tumor was completely duplicated, the tumor with muscularis propria and serosa was resected (Figure 5C–E). The whole tumor was removed from the stomach using a standard endoscope.

## 3. Results

All procedures were elective, and the main steps of the resection were alike in all cases. Mean resection time was 107.5 min, however in case of the tumor with the diameter approximately 40 mm, the procedure was extended by about 30 min. There were no adverse events during or after the procedure (adverse events rate 0%). The length of hospital stay was 3 days. The post-procedural observation period was uneventful. Prophylactic antibiotics were administered before all of procedures. In addition, PPI were used for 6 weeks after the procedure. Technical success was 100% and complete resection was confirmed in post-resection material trough pathologic assessment. Resection margin was R0 with a therapeutic success of 100%. All tumors were confirmed to be GIST with <5 mitoses/50 HPF. Based on GEIS guidelines, risk of progression was very-low, therefore adjuvant therapy and follow-up were not required [21]. (Table 2, Figure 6)

## 4. Discussion

The safety and effectiveness of a hybrid endoscopic technique for gastric GISTs type III, with a diameter between 20 and 40 mm, were evaluated in this study. The optimal treatment of GISTs depends on various factors like the tumor size, presence of metastases and stage of the disease. The additional factor determining further management is the malignancy potential. Despite the malignancy risk estimation supported by pathologic assessment, the resection of tumors significantly increases the chance for cure. Therefore, resection of GISTs is recommended as diagnostic and therapeutic approach [22]. European Society for Medical Oncology (ESMO) and Japanese Society of Clinical Oncology recommend surgical resection GISTs smaller than 20 mm [15,16]. Based on ESMO guidelines, the standard approach to GISTs ≧ 20 mm should consider surgical treatment as a first option. In terms of qualification for complementary management, the tumor location should be taken into account. The presence of the tumor with gastric location deprived metastases is related to a very low risk of progression without indication for adjuvant therapy and follow-up [15].

In our group, all patients were qualified for endoscopic treatment after multidisciplinary consensus. As mentioned before, endoscopic resection methods of gastric GISTs include EBL, ESE, ESD and EFTR. However, considering features of the tumor, techniques included EBL, ESE and ESD were not recommended due to possible incomplete resection. Therefore, EFTR as a resection technique of the tumor with a safe margin, was chosen. This method was firstly presented by Suzuki et al. [23] and further studies have confirmed the effectiveness of this resection modality for tumors infiltrating the MP. In two studies, a total of 59 gastric GISTs were endoscopically removed using EFTR without laparoscopic assistance [24,25]. Technically, for the procedure, an endoscopic snare, clips and an endoloop were used. Moreover, the key to the method was performing a controlled perforation [26]. Despite the high effectiveness with the technical success of 100% and a low risk of other complications, this technique seems advanced and time-consuming. In turn, gFTRD (gastric FTRD; Ovesco Endoscopy, Tübingen, Germany) may be other option in terms of cost and availability. Meier et al. [27] resected six gastric GISTs with a diameter of under 20 mm. Complete R0 resection was achieved in three tumors, with a total technical success 89.7% (26/29 of submucosal lesions). This method seems to be less technically demanding than previously described. However, gFTRD is dedicated to diagnostic or therapeutic full-thickness-resection in the case of GISTs < 20 mm. Huang et al. performed endoscopic full-thickness resection of 13 gastric GISTs with the closure of perforations with metal clips followed by an abdominal paracentesis to decrease the abdominal cavity pressure [20]. For more extensive perforation, the retina mending method is recommended [26]. Moreover, postoperative gastrointestinal decompression, proton pump inhibitors and antibiotics should be administered to prevent postoperative infection [26]. Currently presented variants of gEFTR may not allow for curable resection. In particular, when the significant MP involvement, extraluminal protrusion or infiltration of the whole gastric wall is expected.

Presented method in this manuscript, it consisted of two main steps. The performance of the incision around the tumor, cutting submucosa and closing neighboring gastric walls in the near of the tumor leading to lift up the lesion and provide the full-thickness wall resection with the pathologic lesion. The application of Apollo OverStitch TM set was expanded beyond bariatric indications, such as suturing after endoscopic resection [28]. However, from available tools, Apollo OverStitch TM set allows for bringing neighboring gastric walls together and resect tumors without the necessity of the perforation performance.

There are couple of limitations of this method. Firstly, current procedure may be related with higher costs and the required technical skills of the endoscopic operator. However, the technical success rate was 100% with no complications during or after the procedure. The limitation of complications occurrence may be related to the fact that in this method the perforation is not performed during the resection due to using a suturing system before the knife cutting. Moreover, hospital stay duration is shorter and general costs are lower than after surgical treatment. It is important to note that therapeutic success rates are comparable to standard surgical treatment, especially in the case of low-risk tumors, when lymph nodes resection is not indicated. In terms of procedure duration, our hybrid approach could be comparable with ESD. However, ESD can only be used for GISTs originating from the superficial MP layer. In addition, major complications such as perforation (8.2%) and bleeding (15.6%) and less common such aspiration pneumonia, venous thromboembolism and air embolism should be taken into account [19]. In turn, the duration time of the presented hybrid technique was lower than surgical procedures, both laparoscopic and classical (107.5 min vs. 147.8 ± 59.3 vs. 139.2 ± 62.1, respectively) [29].

Additional, for standardization of this method an increment of case volume is necessary. However, it seems to be an alternative method for patients with gastric GISTs type III and size between 20–40 mm, disqualified from surgical treatment.

## 5. Conclusions

All confirmed GISTs require removal either by standard surgery or alternative endoscopic procedures. There is currently no optimal method to remove gastric GISTs with tumor size of 20 to 40 mm, type III. In the presented hybrid approach patient benefits outweigh its limitations. Due to a small patient population, further studies are necessary to compare treatment modality with other techniques.

## Figures and Tables

**Figure 1 jcm-09-01776-f001:**
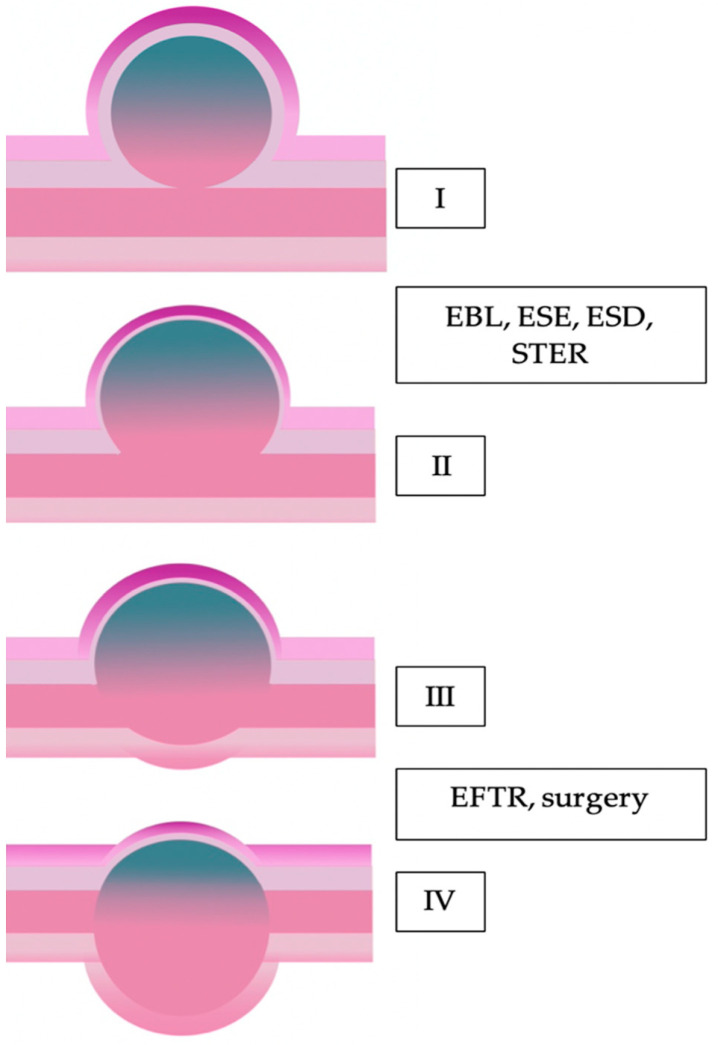
Adapted from Kim et al. Classification of GISTs (gastrointestinal stromal tumors) [7]. Type I—GIST with a very narrow connection with the proper muscle layer and protrudes into the luminal side, like a polyp; Type II—with a wider connection with the proper muscle layer and protrudes into the luminal side at an obtuse angle; Type III—is located in the middle of the gastric wall; Type IV—protrudes mainly into the serosal side of the gastric wall.

**Figure 2 jcm-09-01776-f002:**
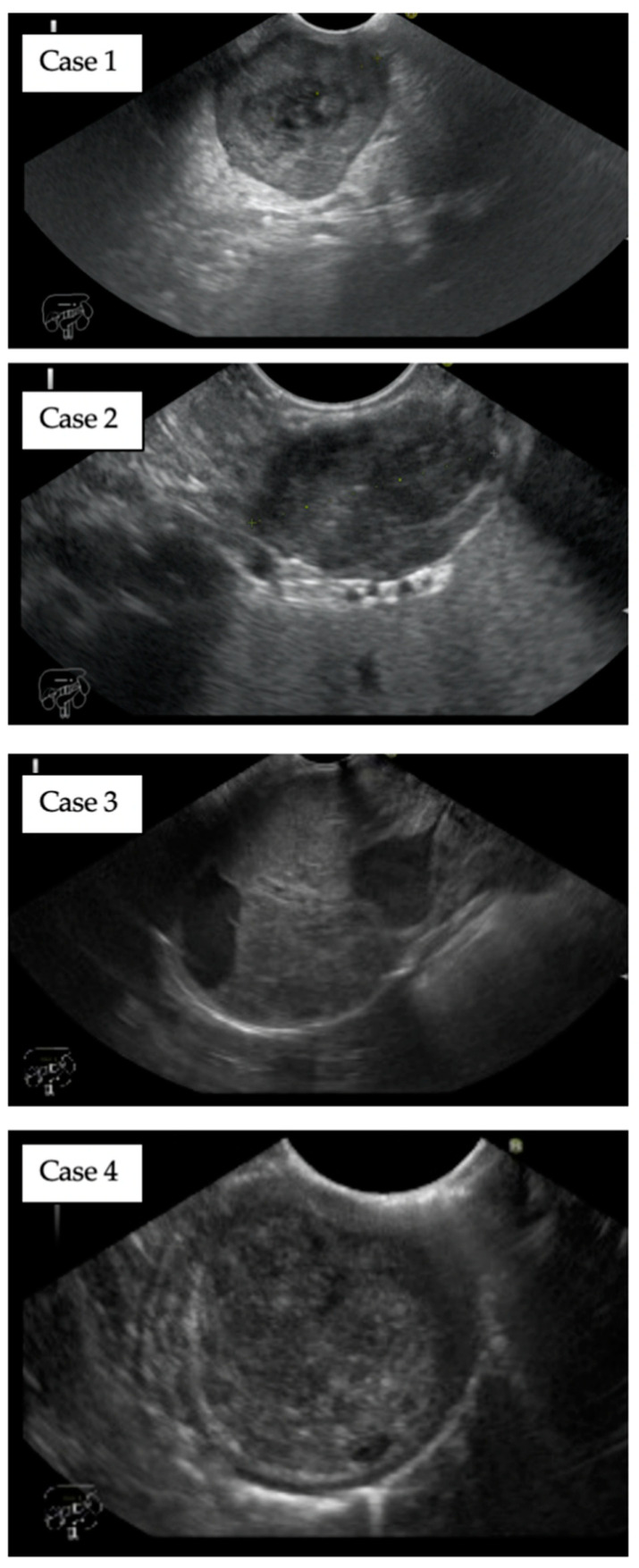
Endoscopic ultrasound evaluation of gastric GISTs. Typical appearance of GIST in endoscopic ultrasound. All tumors of a mixed echogenicity were located in the middle of the gastric wall and have a deep connection with *muscularis propria*. The average size was 28.75 mm.

**Figure 3 jcm-09-01776-f003:**
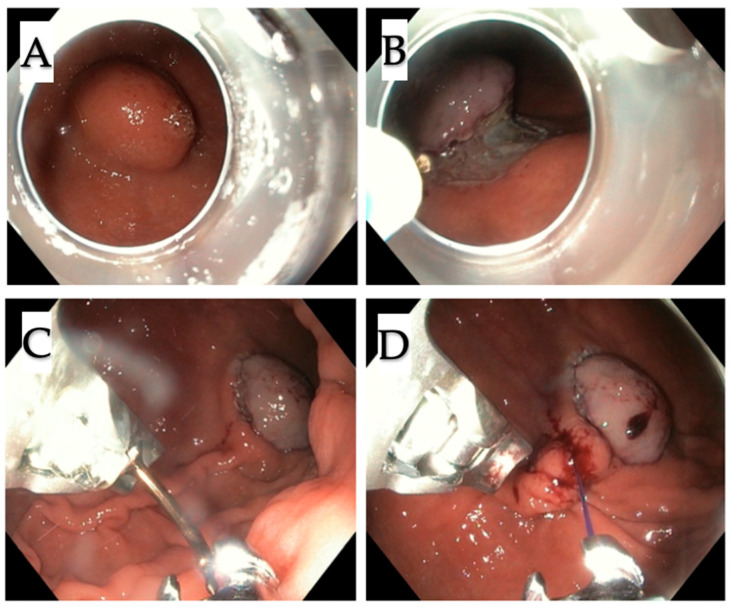
First step of the procedure (case 4). (**A**,**B**) Marking the tumor and circular incision around the tumor with submucosal dissection; (**C**,**D**) then adjacent folds of the stomach wall with Apollo OverStitch were bringing together. Tumor was partly lifted up, which provided protection against the perforation during further resection.

**Figure 4 jcm-09-01776-f004:**
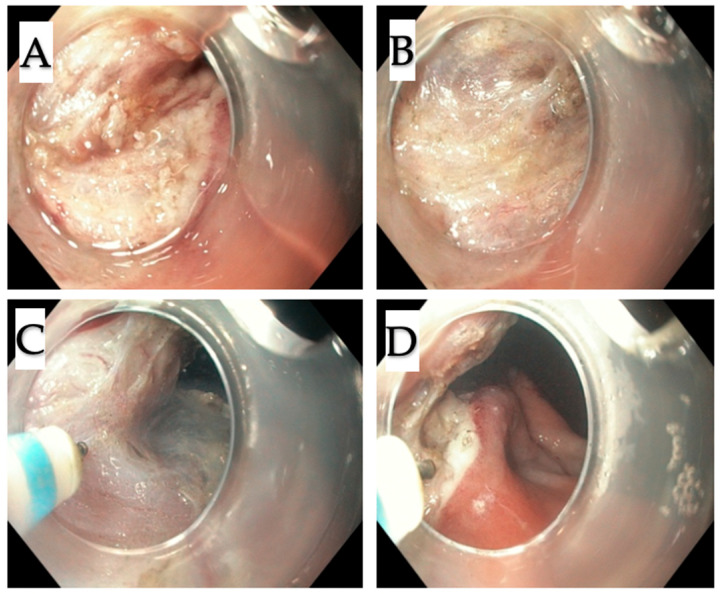
Step two. (**A**–**D**) Subsequent layers below the tumor were cut with the incision of muscularis propria.

**Figure 5 jcm-09-01776-f005:**
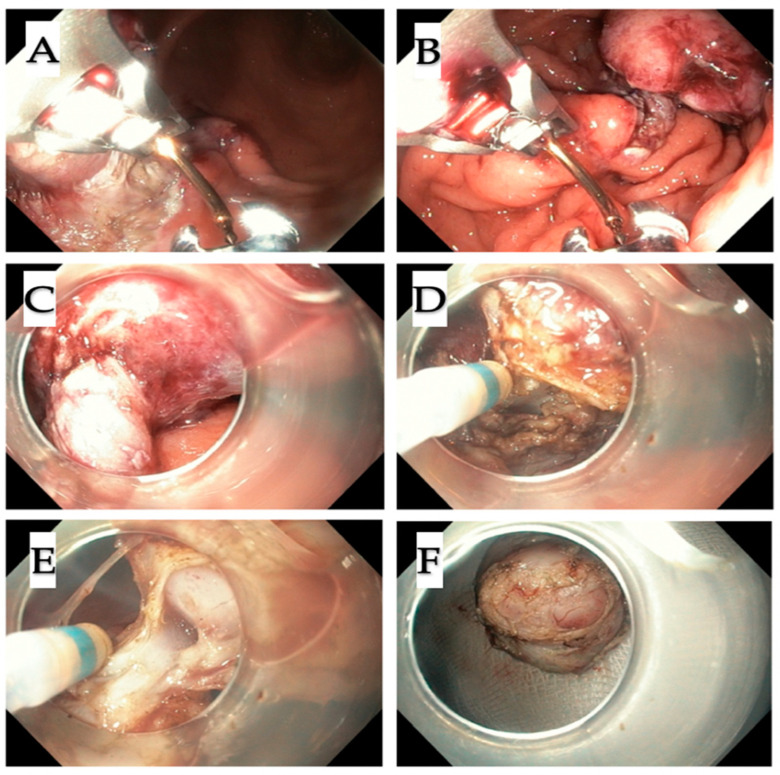
Step three. Step three. (**A**,**B**) After revealing the connection of the tumor with the deep layer of the muscularis propria, the adjacent gastric folds have been again brought together in another site (**C**–**F**) to lift up the resection site, allowed for safe and transmural cut off the tumor; (**A**,**B**) successive sutures were inserted (**C**–**F**) to enucleate it completely from the MP layer, thus the full-wall resection was performed.

**Figure 6 jcm-09-01776-f006:**
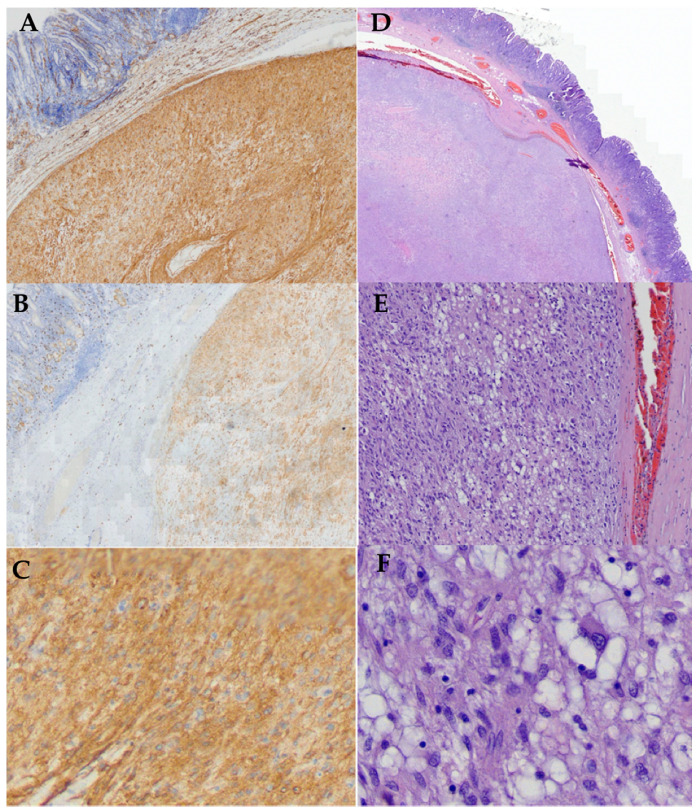
Immunohistochemical markers of the tumors. Membranous and cytoplasmic staining for (**A**) CD 117 (+), CD 34 (+), DOG-1 (+). (Case 1–4); (**B**) CD34 (+) strong positive (4×); (**C**) CD117 (+) mild positive (×4); (**D**) strong and diffuse expression of DOG1 (+), (**D)** membranous and cytoplasmic (2×), (**E**) 10×; (**F**) 40× H&E—mixed epithelioid/spindle cell morphology with subnuclear vacuoles, eosinophilic cytoplasm and abundant myxoid stroma with thin-walled blood vessels.

**Table 1 jcm-09-01776-t001:** Patient characteristics.

No	Age	Gender	Concomitant Disease	Approximate Size of the Tumor (mm)	Location	Type of the Tumor	History of Bleeding from the Tumor
Case 1	62	K	None	20 mm	between fundus and body of the stomach	3	yes
Case 2	65	M	Hypertension	25 mm	between fundus and body of the stomach	3	no
Case 3	68	M	Hypertension	30 mm	antrum	3	yes
Case 4	77	M	Thrombocytopenia, myelodysplastic syndrome	40 mm	middle of the gastric body	3	no

**Table 2 jcm-09-01776-t002:** Results of the treatment, complications, follow-up.

Parameter	Value	
Mean resection time (min)	107.5	
Technical success	ESD tool + Apollo OverStitchFM	4 (100%)
Type of adverse events	none	
Adverse events rate (%)	0	0%
Complete resection; rate (%)	Confirmed in post-resection material (pathologic assessment)	4 (100%)
Resection margin (R); rate (%)	R0	4 (100%)
Therapeutic success	4 (100%)	
Histopathology	Confirmed GIST <5 mitoses / 50 HPF	4 (100%)
NIH risk classification	Very low	4 (100%)
Mean time of hospitalization (days)	3	

Continuous variables were reported using mean ± standard deviation (SD). Categorical variables were reported using proportion (%).

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
