# Peer review of "Optimal Endoscopic Resection Technique for Selected Gastric GISTs. The Endoscopic Suturing System Combined with ESD—a New Alternative?"

_jcm, 2020, doi:10.3390/jcm9061776_

Round 1
Reviewer 1 Report
In this study the Autohors describe th application of an hybrid method (resction plus sutiring) for the ndoscopic rmoval of GISTs 20-40 mm size. The technical feasibility is demostrated although in only 4 patients, the long term efficacy is not provided.
Major points:
1. Although the Authors have pointed out that "..the point is not concerning the feasibility of endoscopic resection...." but "to remove the lesion safely with a significant reduction in the risk of recurrence", no evidence is given related to this point. please give evidence of follow up, comparable to other resective technique, or delete reference to long term efficacy.
2. Due to the limited number of patients treated, I suggest concentrating on the technique rather than on the clinical issues. Therefore, background and discussion must be shortened and refocused.
Author Response
Dear Reviewer,
On behalf of our team, I would like to thank for all of the important comments. We agree with suggestions, therefore the manuscript was changed according to the suggestions.
The changes are visible as a "track" option on file.
Kindly regards
Katarzyna M Pawlak, MD, PhD

Reviewer 2 Report
General: The authors have identified an interesting topic here on “Optimal endoscopic resection technique for selected gastric GISTs. The endoscopic suturing system combined with ESD- a new alternative?” which is an interesting topic.
Some concerns which will need to be addressed are:
The title is appropriate.
Abstract:
Line 22-24: Authors mention “Despite well-known treatment strategies based, among others, on general clinical assessment and the type of tumor according to Miettinen classification, adequate management remains questionable in some cases” – this statement is confusing, cannot understand what the authors are trying to say here. This needs to be rewritten.
Introduction:
Line 45-47: Needs to be changed as mentioned above.
Line 56: “Inasmuch” change to In as much
Line 56-57 “Inasmuch prevalence of SELs was estimated at 0.36% what confirmed the fact,” needs to be rewritten.
Line 89: “[Error! Bookmark not defined.].” needs to be deleted
Line 91-92: authors mention “a. The selection of endoscopic techniques should consider the connection with muscularis propria (MP)…”. Rephrase the sentence please. Consider connection? Or invasion of?
Materials and Methods:
Looks ok
Images look poor quality.
Discussion:
429-431: “A lack of complications might be related to the fact that in this method the perforation is not performed for resection the tumor”…is unclear. Rephrase it.
Limitations should be in one place rather than mentioning it sperately.
Overall a well conducted study.
Major drawback is that the article is poorly written and English is poor. Needs to be checked by a native English speaker and sentences will need to be repharsed.
Author Response
Dear Reviewer,
On behalf of our team, I would like to thank for all of improtant comments. We agree with suggestions, therefore the manuscript was changed ("track option"
Also, the manuscript was checked by a native English speaker.
The size of figures was changed. If necessary, they can be sent in an attachment.
Kindly regards,
Katarzyna M Pawlak, MD, PhD

Round 2
Reviewer 1 Report
The comments have been modified
This manuscript is a resubmission of an earlier submission. The following is a list of the peer review reports and author responses from that submission.
Round 1
Reviewer 1 Report
The authors present a novel means of tackling GISTs. There are several errors in English in the text which need changing such as line 84, line 88, line 95, line 109 etc. Individually these are minor, but together they represent a challenge to the readership and I would recommend a redraft of the document.
For me the main issue is the lack of pertinent description of use of the overstitch - it's not clear to me what the authors mean by duplication of the muscle layer. The pictures are beautiful, but again fail to adequately explain the nature of the intervention
I would recommend this section is rewritten to make it clearer for readers
Author Response
Dear Reviewer,
We are so grateful and appreciate all comments.
We made all suggested corrections, especially with clarifying of the procedure (step by step) and usefulness of Apollo OverStitch in this type of tumors. We added two scheme with the procedure description and we changed Figure for better understanding the point of procedure. Also, our paper was corrected by english native speaker.
We hope it will be satisfactionary.
With best wishes
KMP

Reviewer 2 Report
This manuscript evaluated the availability of a new hybrid resection technique for gastric GIST. The authors performed non-exposed endoscopic full-thickness resection of gastric GISTs combined with suturing with the Apollo OverStitch System for four patients, and technical and therapeutic success was achieved in all four cases.
As this article presented a new minimally-invasive technique for the resection of gastric GISTs, it may be a new alternative treatment for gastric GISTs. Therefore, the results will be of interest to clinicians in the field.
However, the following major and minor issues require clarification:
Major
- The authors should show the characteristics of each case as a table instead of Table 1.
- Figures and Figure legends should be improved. First, the authors should state clearly in the Figure legends which case they selected and what each of the endoscopic images showed. Second, Figure 2-5 included too many images. Please reduce the number of images. Third, Figure legend 2-4 should be summarized.
- The authors should introduce this technique using some easily understood schematic illustrations.
Minor
- The authors should replace the order of “Figure 5” with “Figure 6”.
I hope these comments will be helpful for improving this manuscript.
Author Response
Dear Reviewer,
We are so grateful and appreciate all comments.
We made all suggested corrections, especially with clarifying of the procedure, step by step and usefulness of Apollo OverStitch in this type of tumors. We added two scheme with the procedure description and we changed Figure for better understanding the point of procedure. We changed the table as was suggested. Also, our paper was corrected by english native speaker.
We hope that all corrections are satisfactory.
Best regards
KMP

Round 2
Reviewer 2 Report
The manuscript is considerably improved. However, the authors have not yet reconsidered several issues I had pointed out.
- (Figure 2, 4-6) Please state clearly in the Figure legends which case they selected. It should be easy to number the cases in Table 1 and refer the case numbers in Figure legends (e.g. Case 1).
- (Figure 2, 5) Please number the images (e.g. Figure 2A-D) and state clearly in the Figure legends what each of the images showed.
- The authors should replace the order of “Figure 5” with “Figure 6”.
Author Response
Review-round 2
Thank you again for important comments and suggestions. All changes are red, also "track changes" option has been applied.
1.(Figure 2, 4-6) Please state clearly in the Figure legends which case they selected. It should be easy to number the cases in Table 1 and refer the case numbers in Figure legends (e.g. Case 1).
All figures (2,4-6) with Table 1 have been corrected according to the Reviewer's recommendations.
2.(Figure 2, 5) Please number the images (e.g. Figure 2A-D) and state clearly in the Figure legends what each of the images showed.
Figure 2 and 5 also have been changed as the Reviewer recommended.
3.The authors should replace the order of “Figure 5” with “Figure 6”.
We replaced Figure 5 with 6.
Best regards
KM Pawlak
